# Origin of a folded repeat protein from an intrinsically disordered ancestor

Hongbo Zhu, Edgardo Sepulveda, Marcus D Hartmann, Manjunatha Kogenaru[†], Astrid Ursinus, Eva Sulz, Reinhard Albrecht, Murray Coles, Jörg Martin, Andrei N Lupas*

Department of Protein Evolution, Max Planck Institute for Developmental Biology, Tübingen, Germany

**Abstract** Repetitive proteins are thought to have arisen through the amplification of subdomain-sized peptides. Many of these originated in a non-repetitive context as cofactors of RNA-based replication and catalysis, and required the RNA to assume their active conformation. In search of the origins of one of the most widespread repeat protein families, the tetratricopeptide repeat (TPR), we identified several potential homologs of its repeated helical hairpin in non-repetitive proteins, including the putatively ancient ribosomal protein S20 (RPS20), which only becomes structured in the context of the ribosome. We evaluated the ability of the RPS20 hairpin to form a TPR fold by amplification and obtained structures identical to natural TPRs for variants with 2–5 point mutations per repeat. The mutations were neutral in the parent organism, suggesting that they could have been sampled in the course of evolution. TPRs could thus have plausibly arisen by amplification from an ancestral helical hairpin.

*For correspondence: andrei. lupas@tuebingen.mpg.de

Present address: [†]Department of Life Sciences, Imperial College London, London, United Kingdom

Competing interests: The authors declare that no competing interests exist.

## Introduction

Most present-day proteins arose through the combinatorial shuffling and differentiation of a set of domain prototypes. In many cases, these prototypes can be traced back to the root of cellular life and have since acted as the primary unit of protein evolution (*Anantharaman et al., 2001*; *Apic et al., 2001*; *Koonin, 2003*; *Kyrpides et al., 1999*; *Orengo and Thornton, 2005*; *Ponting and Russell, 2002*; *Ranea et al., 2006*). The mechanisms by which they themselves arose are however still poorly understood. We have proposed that the first folded domains emerged through the repetition, fusion, recombination, and accretion of an ancestral set of peptides, which supported RNA-based replication and catalysis (the RNA world *Bernhardt, 2012*; *Gilbert, 1986*) (*Alva et al., 2015*; *Lupas et al., 2001*; *Söding and Lupas, 2003*). Repetition would have been a particularly prominent mechanism by which these peptides yielded folds; six of the ten most populated folds in the Structural Classification of Proteins (SCOP) (*Murzin et al., 1995*) – including the five most frequent ones – have repetitive structures. In all cases, their amplification from subdomain-sized fragments can also be retraced at the sequence level in at least some of their members.

One of these highly populated repetitive folds is the αα-solenoid (SCOP a.118), whose most widespread superfamily is the tetratricopeptide repeat (TPR; a.118.8). This was originally identified as a repeating 34 amino-acid motif in Cdc23p of *Saccharomyces cerevisiae* (*Sikorski et al., 1990*) – hence its name. Since then, TPR-containing proteins have been discovered in all kingdoms of life, where they mediate protein-protein interactions in a broad range of biological processes, such as cell cycle control, transcription, protein translocation, protein folding, signal transduction and innate immunity (*Cortajarena and Regan, 2006*; *Dunin-Horkawicz et al., 2014*; *Katibah et al., 2014*; *Keiski et al., 2010*; *Kyrpides and Woese, 1998*; *Lamb et al., 1995*; *Sikorski et al., 1990*). The first crystal structure of a TPR domain (*Das et al., 1998*) showed that the repeat units are helical hairpins,

**eLife digest** All life is built upon the chemical activity of proteins. For this activity, proteins need to fold into specific 3D structures. Protein folding is complicated and easily disrupted, and its evolutionary origin remains poorly understood. A possibility is that folded proteins arose through different genetic processes from shorter pieces of protein called peptides, which participated in an ancient, primordial form of life. One of these processes involves the same peptide being repeated within one protein chain.

In 2015, researchers identified 40 primordial peptides whose sequences appear in seemingly unrelated proteins. The study suggested that repetition allows peptides that are unable to fold by themselves to yield folded proteins. Now, Zhu et al. – who are members of the same research group who performed the 2015 study – have explored experimentally whether one of these peptides could indeed yield a folded protein by repetition.

The studied primordial peptide gave rise to several protein folds seen today, including – by repetition – a type of fold called TPR. Zhu et al. tried to retrace the emergence of the TPR fold by taking a descendant of the primordial peptide from a ribosomal protein, which is unable to fold without the assistance of an RNA scaffold, and repeating it three times within the same protein chain. The ribosome is a central component of all living cells and evolves very slowly, and so the peptide Zhu et al. took from it is likely to retain many properties of its primordial ancestor.

Further experiments found that the repeated peptide was indeed able to fold into a TPR-like structure, but needed several mutations to do so. Introducing these mutations back into the ribosomal protein, however, did not affect the survival and growth of the cell. Thus, they could have occurred without adverse effects during evolution.

Structure is a prerequisite for chemical activity, but it is activity that is under selection in living beings. Having produced a new protein, Zhu et al. will now explore ways of endowing it with a selectable activity.

stacked into a continuous, right-handed superhelical architecture with an inner groove that mediates the interaction with target proteins (*Forrer et al., 2004*). The hairpins interact via a specific geometry involving knobs-into-holes packing (*Crick, 1953*) and burying about 40% of their surface between repeat units. This tightly packed, superhelical arrangement of a repeating structural unit is typical of all αα-solenoid proteins (*Di Domenico et al., 2014*; *Kajava, 2012*; *Kobe and Kajava, 2000*).

Comparison of TPRs from a variety of proteins reveals a high degree of sequence diversity, with conservation observed mainly in the size of the repeating unit and the hydrophobicity of a few key residues (*D'Andrea and Regan, 2003*; *Magliery and Regan, 2004*). Nevertheless, almost all known TPR-containing proteins can be detected using a single sequence profile (*Karpenahalli et al., 2007*), underscoring their homologous origin. As their name implies, TPR proteins generally contain at least two unit hairpins in a repeated fashion. The few that have only one hairpin, notably the mitochondrial import protein Tom20 (*Abe et al., 2000*), are clearly not ancestral based on their phylogenetic distribution and functionality, implying that the ancestor of the superfamily already had a repeated structure. In searching for the origin of TPRs, we hypothesized that the hairpin at the root of the fold might either have been part of a different, non-repetitive fold or have given rise to both repetitive and non-repetitive folds at the origin of folded domains. Either way we hoped that we might find α-hairpins in non-repetitive proteins that are similar in both sequence and structure to the TPR unit, suggesting a common origin. Here we show that such hairpins are detectable and that one of them, from the ribosomal protein RPS20 (*Schluenzen et al., 2000*), can be customized to yield a TPR fold by repetition, with only a small number of point mutations that are neutral for the parent organism. Ribosomal proteins most likely constitute some of the oldest proteins observable today and are still intimately involved in an RNA-driven process: translation (*Fox, 2010*; *Hsiao et al., 2009*). They are mostly incapable of assuming their folds outside the ribosomal context (*Peng et al., 2014*) and thus belong to a class of intrinsically disordered proteins that become structured upon binding to a macromolecular scaffold (*Dyson and Wright, 2005*; *Habchi et al., 2014*; *Oldfield and Dunker, 2014*;

*Peng et al., 2014*; *Varadi et al., 2014*). This hairpin therefore plausibly retains today many of the properties likely to have been present in the ancestral peptide that gave rise to the TPR fold.

## Results and discussion

### Recently amplified TPR arrays in present-day proteins

Repetitive folds with variable numbers of repeats, such as HEAT, LRR, TPR or β-propellers, usually have some members with a high level of sequence identity between their repeat units (*Dunin-Horkawicz et al., 2014*). In these proteins, the units are more similar to each other than to any other unit in the protein sequence database, showing that they were recently amplified. In a detailed study of β-propellers (*Chaudhuri et al., 2008*), we found that this process of amplification and differentiation has been ongoing since the origin of the fold. TPR proteins show a similar evolutionary history. In some proteins, most of the repeats can be seen to have been amplified separately and to a different extent in each ortholog, pointing to their recent origin (*Figure 1a*); in others, the amplification must have occurred much earlier, as their ancestor already had fully differentiated repeats (*Figure 1b*). In recently amplified proteins, such as the ones shown in *Figure 1a*, within which repeats frequently have >80% pairwise sequence identity, tracking the probable α-hairpin at the root of the amplification is a fairly straightforward proposition. We wondered, however, whether it might be possible to go much further back in time and track the original α-hairpin from which the first TPR protein was amplified. We therefore searched for TPR-like α-hairpins in non-repetitive proteins as present-day descendants of the original hairpin.

### Identification of helical hairpins resembling the TPR unit

We had previously developed a profile-based method, named TPRpred, specially designed for the detection of TPRs and related repeat proteins with high sensitivity from sequence data (*Karpenahalli et al., 2007*). Here, in a first step, we used TPRpred to scan protein sequences in the Protein Data Bank (PDB) (*Berman et al., 2000*) for peptides that share statistically significant similarity to the TPR sequence profile and yet have not been annotated as TPR in Pfam (*Finn et al., 2014*); we used a p-value cutoff = 1.0e−4, which leads to an estimated false discovery rate of 1.0%, see Materials and methods. We ignored tandem repeats in the hit list and focused only on the singleton cases. Subsequently, we compared the structures of these helical hairpin singletons to the average TPR hairpin and removed non-hairpin-like structures. This yielded 31 helical hairpins that are similar to the TPR unit with respect to both sequence and structure. Among them, 22 are part of solenoid-like structures and were discarded. The remaining nine hits belong to three families: (I) mitochondrial import receptor subunit Tom20; (II) microtubule interacting and transport (MIT) domain including katanin (*Iwaya et al., 2010*); and (III) 30S ribosomal protein S20 (RPS20) (*Figure 2*).

The similarity of Tom20 and MIT domains to TPR proteins has been noted before (*Abe et al., 2000*; *Iwaya et al., 2010*; *Scott et al., 2005*), but the similarity of RPS20 was surprising and drew our attention particularly due to the ancestrality attributed to ribosomal proteins. To further explore the similarity between the helical hairpin in RPS20 (in short, RPS20-hh) and TPR, we used TPRpred to rank the RPS20 sequences in Pfam (*Finn et al., 2014*). The top-scoring hit was RPS20-hh from *Thermus aquaticus* (NCBI accession number = WP_003044315.1, UniProt id = B7A5L8_THEAQ), which matches the TPR unit sequence profile at a p-value of 5.4e−07, almost an order of magnitude better than the second hit (see *Supplementary file 1D*). Furthermore, we examined the surface residues of RPS20-hh fragments to assess their suitability to occur in a tandem repeat mode, as in TPRs. To this end, we first defined five interface positions on the TPR helical hairpin and transferred the definition to RPS20-hh according to their structure alignment (positions 3, 7, 10, 21 and 28 using TPR unit numbering). Then, we searched for RPS20-hhs with as many hydrophobic residues as possible at these interface positions. We found 42 RPS20-hhs that contain at least three hydrophobic residues out of the five interface positions. Among them, the only RPS20-hh predicted to match the TPR unit profile above a p-value of 1.0e−4 was again the RPS20 from *T. aquaticus*, in which three of the five interface residues are hydrophobic (L10, I21 and V28). We therefore chose this helical hairpin (RPS20-hhta) to construct a TPR-like solenoid by amplification (*Figure 3*).

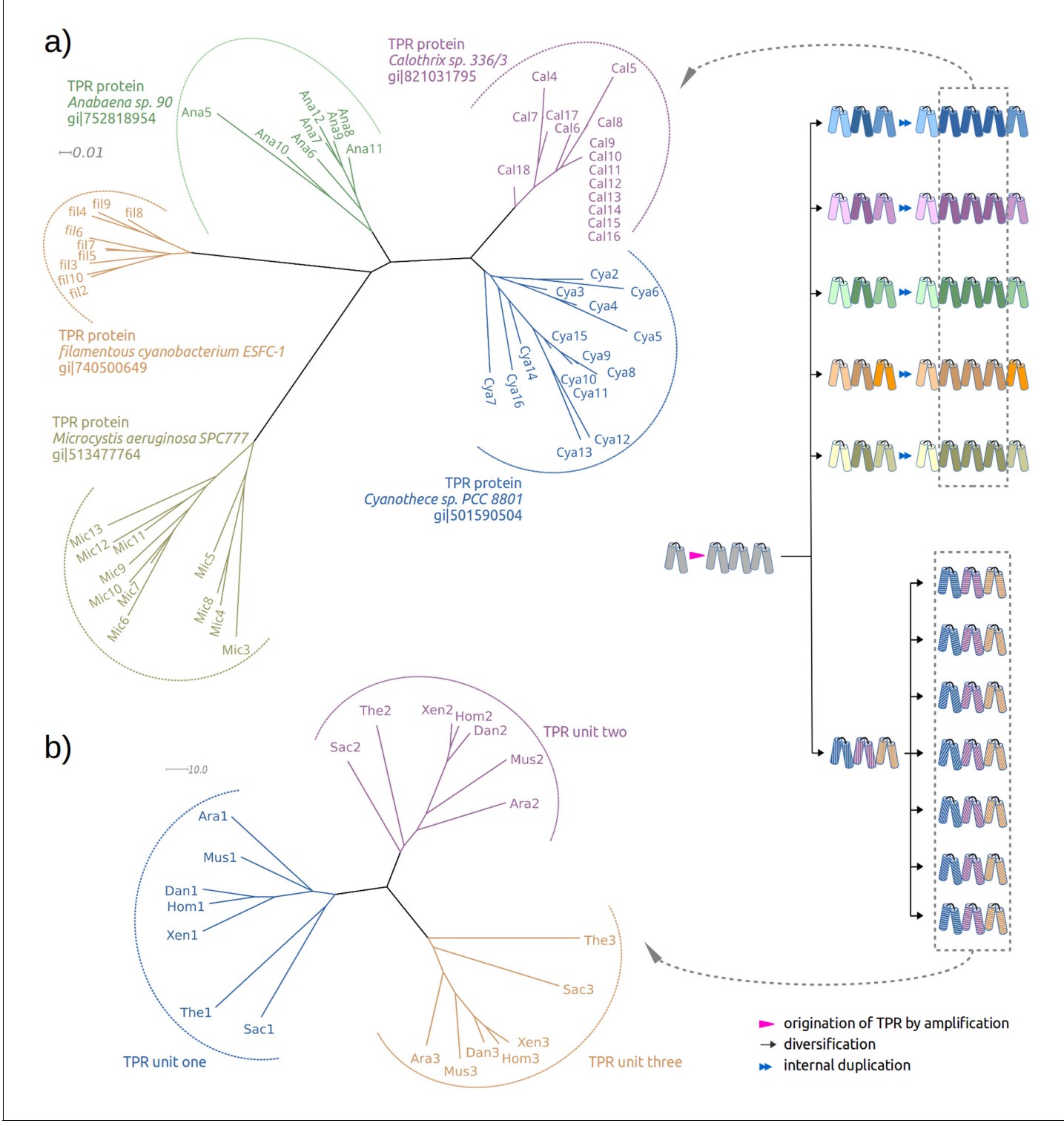

**Figure 1.** Two evolutionary scenarios for TPRs, illustrated by neighbor-joining phylogenetic trees. (a) Amplification from single helical hairpin, as seen in TPR proteins from Cyanobacteria. (b) Divergent evolution of a TPR with multiple repeat units, as seen in the TPR domains of Serine/threonine-protein phosphatase 5 (Ara: *Arabidopsis thaliana*, Dan: *Danio rerio*, Hom: *Homo sapiens*, Mus: *Musca domestica*, Sac: *Saccharomyces cerevisiae*, The: *Theileria annulata*, Xen: *Xenopus (Silurana) tropicalis*). Since evolutionary reconstructions are subject to Occam's razor and reflect the hypothesis with the fewest assumptions, we have postulated here one amplification event from one precursor hairpin. Our findings would however also be fully compatible with the precursor hairpin yielding a population of homologous variants, some of which were independently amplified to TPR-like folds; one or more survivors among these would have become the ancestor(s) of today's TPR proteins. In this more complex scenario, the homology of TPR proteins, which

*Figure 1 continued on next page*

*Figure 1 continued*
we trace through the comparison of individual hairpins, is still given, but the TPR fold could have arisen from several independent amplifications, and not just a single one.
The following figure supplements are available for figure 1:

**Figure supplement 1.** Multiple sequence alignments of recently amplified TPR repeat units.
**Figure supplement 2.** Multiple sequence alignments of the three TPR repeat units in serine/threonine-protein phosphatase 5 from seven taxa.

## Design of a TPR array from a RPS20

We focused on the construction of three-repeat TPRs, which represent the most common form of this fold (*D'Andrea and Regan, 2003*; *Sawyer et al., 2013*). For instance, 18 of the 54 non-identical TPR domains in the extended Structural Classification of Proteins database (SCOPe v2.05) (*Fox et al., 2014*) have three repeats. A previously designed three-repeat TPR protein, CTPR3, was also demonstrated to be highly stable, even more so than natural three-repeat TPR proteins (*Main et al., 2003b*). We concatenated three copies of RPS20-hhta as an initial construct, connected by the TPR consensus loop sequence (DPNN). We annotate the two helices in each repeat unit as helix $Ai$ and $Bi$, where $i$ is the index of the repeat unit ($i = 1, 2$ or $3$) (*Figure 3*). Under the hypothesis of common descent between TPR and RPS20 from the same ancestral peptide and retention of ancestral features in RPS20, this basic construct would fold as a TPR solenoid with a minimal number of mutations, ideally none.

When we experimentally made a construct containing no mutations (M0, *Table 1*), it was soluble but remained unfolded under all conditions tested (see Section 2.4). We therefore introduced point mutations into the sequence of RPS20-hhta, aimed at favoring the target structure. Here, we followed the principle of consensus design (*Forrer et al., 2004*; *Main et al., 2003a*), which requires the mutation positions to be occupied by the most commonly observed residues in homologous proteins (*Forrer et al., 2004*). Consensus design methods have been successful in engineering several different repeat proteins with solenoid folds, including ankyrin repeats (*Binz et al., 2003*; *Kohl et al., 2003*; *Mosavi et al., 2002*), TPRs (*Doyle et al., 2015*; *Kajander et al., 2007*; *Main et al., 2003b*), pentatricopeptide repeats (PPRs) (*Coquille et al., 2014*; *Shen et al., 2016*) and leucine rich repeats (*Rämisch et al., 2014*; *Stumpp et al., 2003*). Following these principles, four different sites of mutation (L4W, K7L/R, V9N, I23D/Y, see *Figure 4*) were considered to improve interface hydrophobicity or preserve coevolved positions observed in TPRs (*Sawyer et al., 2013*) (see Materials and methods). Furthermore, as natural TPR proteins tend to exhibit zero net charge (*Magliery and Regan, 2004*), four positively charged residues were also targeted (K2E, K6N, K22E, R25Q/E, see *Figure 4*). This resulted in a set of eight candidate mutation sites. In order to preserve the character of the RPS20-hhta sequence, we restricted the number of mutations in any repeat unit to be at most five.

In most TPR proteins, there is an α-helix at the C-terminus, which interacts with the last TPR unit by covering the hydrophobic surface. This so-called C-terminal 'stop helix' had been observed in all known TPR structures and was considered essential for the solubility of natural TPR proteins (*D'Andrea and Regan, 2003*; *Das et al., 1998*; *Main et al., 2003b*). Most other designed TPRs employ purpose-designed stop helix sequences. Here, we chose to use the RPS20 C-terminal helix to become a natural stop helix, since it is already known to interact favorably with RPS20-hhta (*Figure 3*). Further, we inserted two residues (Asn-Ser) before the first TPR unit as an N-terminal cap to the first helix (*Aurora and Rose, 1998*; *Kumar and Bansal, 1998*), in analogy to a previously designed idealized TPR protein, CTPR3 (*Main et al., 2003b*).

To model the structure of the designed proteins in silico, we fused two structures to create a hybrid template: We used CTPR3 (PDB id: 1na0 chain A) as the structural template for the three RPS20-hhta fragments, and the best-resolved RPS20 structure (PDB id: 2vqe chain T; 2.5 Å) for helix B3 and the stop helix. We built structural models on this hybrid template and tested a variety of mutants using the Rosetta programs *fixbb* and *relax*, which perform fixed-backbone design and structural refinement (*Das and Baker, 2008*; *Doyle et al., 2015*; *Park et al., 2015*;

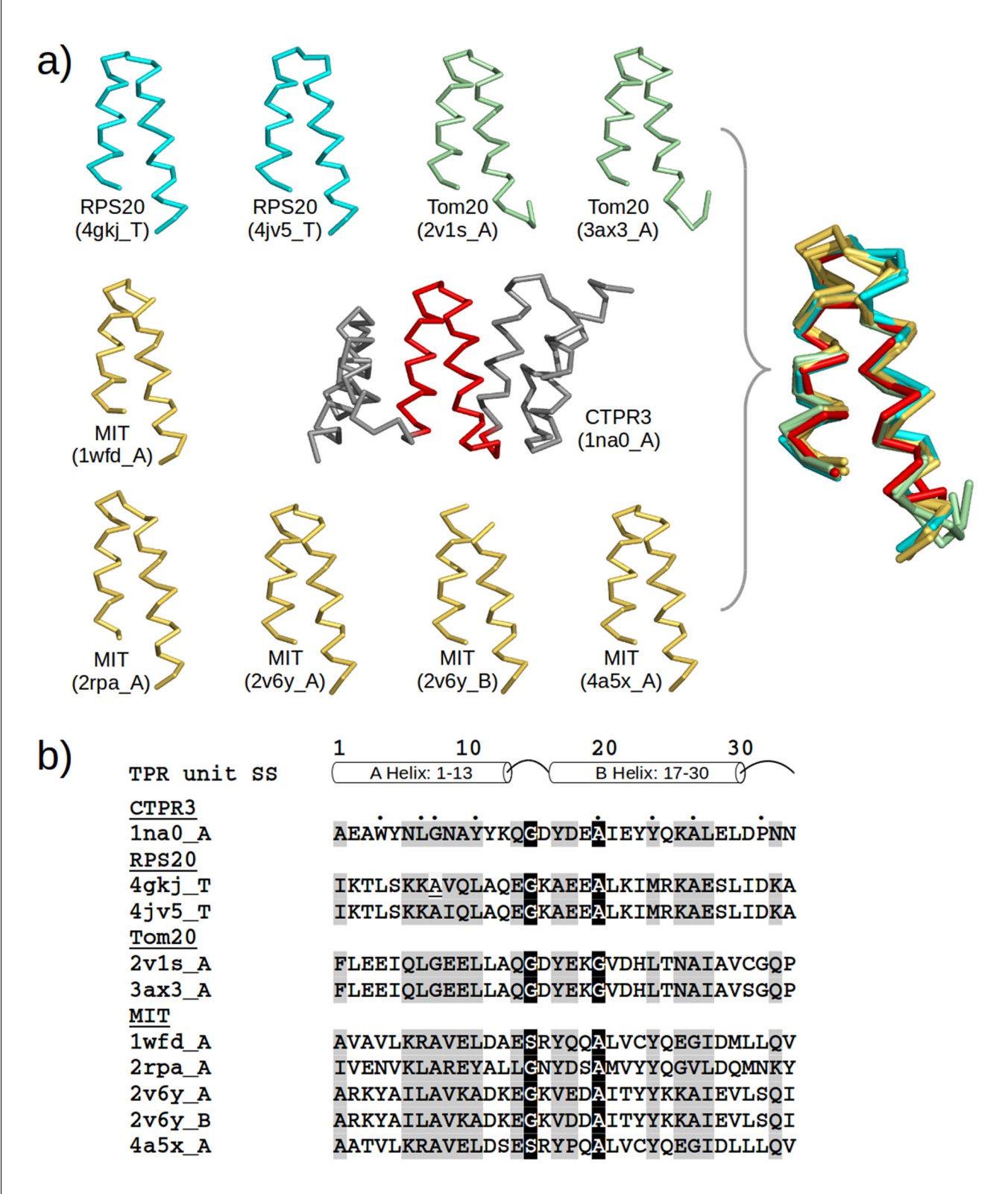

**Figure 2.** TPR-like hairpins found in non-repetitive proteins in the PDB. (a) Structure gallery of non-repetitive helical hairpins in the PDB that share both sequence and structure similarity to TPR unit hairpin. Only the 34 amino-acid helical hairpins are shown. The helical hairpins in 30S ribosomal protein s20 (RPS20), mitochondrial import receptor subunit (Tom20), and microtubule interacting and transport domain (MIT) are depicted in cyan, green, and yellow, respectively. The structure of a TPR with a consensus sequence, CTPR3, is shown in the center with the middle TPR unit highlighted in red. PDB
*Figure 2 continued on next page*

*Figure 2 continued*

IDs and chain names of the proteins are given in parentheses. In the superposition, all helical hairpins are superimposed onto the middle TPR unit of CTPR3. (**b**) Multiple sequence alignment of the helical hairpin sequences listed in (**a**). The eight TPR signature positions are marked by dots in CTPR3. Columns with sequence identity $\geq$ 80% are in black, and columns with sequence identity $\geq$ 50% are in gray.

*Parmeggiani et al., 2015*). The Rosetta energy score of the models calculated for all mutants is depicted in a boxplot (*Figure 4—figure supplement 2*). Among them, five were selected for further testing in vitro (see Materials and methods). These five tested mutants are termed M2, M4E, M4N, M4RD and M5. Their primary structures are listed in *Table 1*.

## Biophysical characterization of designed TPRs and RPS20

We cloned the five TPR designs plus the unmutated construct M0 into pET vectors for expression in *Escherichia coli*. Three proteins (M0, M4RD and M5) could be purified from soluble extracts; the other constructs were insoluble and were refolded from inclusion bodies. In far UV circular dichroism (CD) spectra, all proteins displayed a strong alpha-helical pattern, except M0 and M4RD, which appeared to be unfolded, but not prone to aggregation and precipitation, even at high concentrations. When we studied the melting curves, M4N showed cooperative unfolding with a $T_m$ of 77°C (*Supplementary file 1F*), while the unfolding of M2, M4E and M5 did not conform to a classical two-state transition, consistent with an unstable molten globule-like state. On the other hand, non-cooperative unfolding processes have been demonstrated for perfectly stable TPR repeats and suggested to be common for various types of repeat proteins (*Cortajarena and Regan, 2006*; *Kajander et al., 2007*; *Stumpp et al., 2003*). To clarify this point, urea-induced unfolding transitions were monitored by CD. Like M4N, the three variants M2, M4E and M5 yielded typical cooperative denaturation curves, indicative of folded polypeptides (*Figure 5—figure supplement 2*). The $\Delta G_{U-F}^{H2O}$ values agree well with those reported for other designed TPRs (*Supplementary file 1F*) (*Main et al., 2005*). In line with these findings, M5, the only protein containing tryptophan residues, had a $\lambda_{max}$ of 336 nm in fluorescence emission spectra, as expected for partially shielded aromatic residues. We conclude that four of the five designed TPR variants, M2, M4E, M4N and M5, result in well-folded repeat proteins. To determine the oligomeric state of our folded proteins, we performed static light scattering experiments. Surprisingly, all four constructs were exclusively dimers (*Supplementary file 1F*).

We also examined the ribosomal parent protein RPS20. Within the ribosome, RPS20 is partially embedded in the 16S rRNA, making many nucleic acid contacts. Like many other ribosomal proteins, it is not expected to adopt a stable structure in isolation. Indeed, it has a biased amino acid

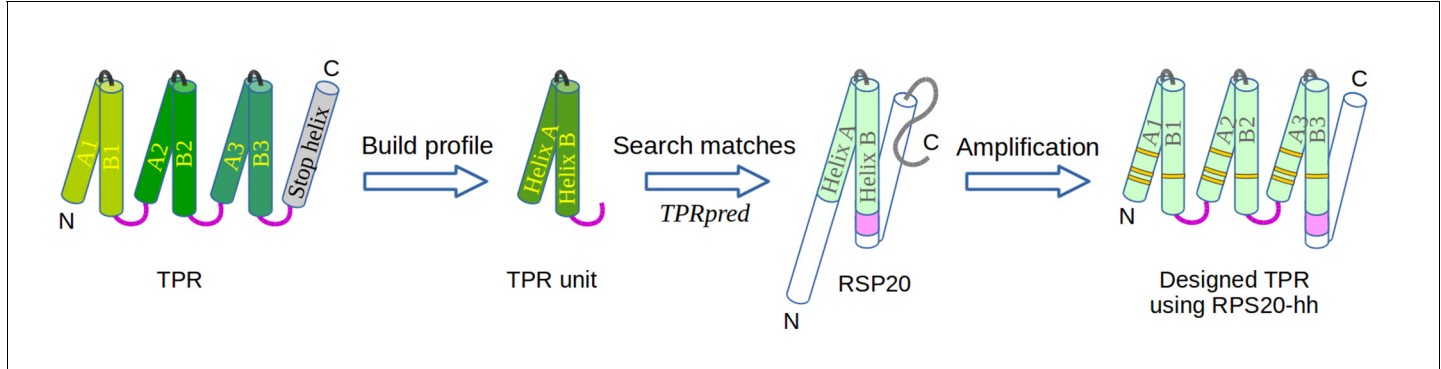

**Figure 3.** The design of TPR using RPS20. RPS20-hh is identified by TPRpred to match the sequence profile of TPR units. Their structures are also very similar (helices are shown as cylinders), except for the last four residues (colored in light and dark magenta). We designed a TPR protein using a RPS20-hh with up to five mutations (yellow strips) in each repeat unit. The C-terminal loop in the TPR unit (dark magenta loop) is used to replace the corresponding C-terminus (light magenta cylinder) of RPS20-hh to connect adjacent repeats. The C-terminal helix in RPS20 (white cylinder) was used as the stop helix in the design.

**Table 1.** The primary structures of the six designed proteins using RPS20-hhta tested in vitro. Point mutations introduced into RPS20-hhta are shown in bold and underlined. The C-terminal four residues in RPS20-hhta were replaced by the consensus loop sequence DPNN in TPRs (underlined). The sequence of the stop helix is italicized. M4NΔC is M4N without stop helix.

| Name | Mutations | Sequence |
|---|---|---|
| M0 | - | NS<br>IKTLSKKAVLLAQEGKAEEAIKIMRKAVSL<u>DPNN</u><br>IKTLSKKAVLLAQEGKAEEAIKIMRKAVSL<u>DPNN</u><br>IKTLSKKAVLLAQEGKAEEAIKIMRKAVSLIDKA<br>*AKGSTLHKNAAARRKSRLMRKVQKL* |
| M2 | K7L, I23Y | NS<br>IKTLSK**L**AVLLAQEGKAEEAIK**Y**MRKAVSL<u>DPNN</u><br>IKTLSK**L**AVLLAQEGKAEEAIK**Y**MRKAVSL<u>DPNN</u><br>IKTLSK**L**AVLLAQEGKAEEAIK**Y**MRKAVSLIDKA<br>*AKGSTLHKNAAARRKSRLMRKVQKL* |
| M4E | K2E, K7L, V9N, I23Y | NS<br>I**E**TLSK**LA**N**L**AQEGKAEEAIK**Y**MRKAVSL<u>DPNN</u><br>I**E**TLSK**LA**N**L**AQEGKAEEAIK**Y**MRKAVSL<u>DPNN</u><br>I**E**TLSK**L**AVLLAQEGKAEEAIK**Y**MRKAVSLIDKA<br>*AKGSTLHKNAAARRKSRLMRKVQKL* |
| M4N | K6N, K7L, V9N, I23Y | NS<br>IKTLS**NL**A**N**LLAQEGKAEEAIK**Y**MRKAVSL<u>DPNN</u><br>IKTLS**NL**A**N**LLAQEGKAEEAIK**Y**MRKAVSL<u>DPNN</u><br>IKTLS**NL**AVLLAQEGKAEEAIK**Y**MRKAVSLIDKA<br>*AKGSTLHKNAAARRKSRLMRKVQKL* |
| M4RD | K2E, K7R, V9N, I23D | NS<br>I**E**TLSK**RA**N**L**LAQEGKAEEAIK**D**MRKAVSL<u>DPNN</u><br>I**E**TLSK**RA**N**L**LAQEGKAEEAIK**D**MRKAVSL<u>DPNN</u><br>I**E**TLSK**R**AVLLAQEGKAEEAIK**D**MRKAVSLIDKA<br>*AKGSTLHKNAAARRKSRLMRKVQKL* |
| M5 | K2E, L4W, K7L, V9N, I23Y | NS<br>I**E**TLSK**LA**N**L**LAQEGKAEEAIK**Y**MRKAVSL<u>DPNN</u><br>I**ETW**SK**LA**N**L**LAQEGKAEEAIK**Y**MRKAVSL<u>DPNN</u><br>I**ETW**SK**L**AVLLAQEGKAEEAIK**Y**MRKAVSLIDKA<br>*AKGSTLHKNAAARRKSRLMRKVQKL* |
| M4NΔC | K6N, K7L, V9N, I23Y | NS<br>IKTLS**NL**A**N**LLAQEGKAEEAIK**Y**MRKAVSL<u>DPNN</u><br>IKTLS**NL**A**N**LLAQEGKAEEAIK**Y**MRKAVSL<u>DPNN</u><br>IKTLS**NL**AVLLAQEGKAEEAIK**Y**MRKAVSLIDKA<br>*AK* |

composition and is predicted to be largely unstructured by many prediction programs (*Figure 4—figure supplement 1*, see also *Supplementary file 1J*). It had been shown previously that isolated RPS20 exhibits only one third helical content by CD (*Paterakis et al., 1983*). For *Thermus* RPS20 specifically, simulations predict a flexible conformation in solution (*Burton et al., 2012*). We cloned RPS20 from *T. aquaticus* and its close relative *T. thermophilus*. Upon expression, both proteins were insoluble and had to be refolded. In static light scattering measurements, both proteins behaved as monomers (*Supplementary file 1F*). Based on CD spectra, which showed a high proportion of random structure, and the absence of defined melting and urea-denaturation curves (*Supplementary file 1F*), we conclude that RPS20 indeed exhibits considerable conformational variation in solution.

## Structure of a designed TPR

To obtain high-resolution structural information on our designed proteins, we set up crystallization trials for all four folded constructs. We obtained crystals and solved the structure of M4N to a resolution of 2.2 Å (*Figure 5a*). The asymmetric unit (ASU) contains three polypeptide chains of almost identical structure (all pairwise $C_\alpha$ RMSD values below 1.4 Å). Notably, all three chains exhibit the desired TPR architecture with three repetitive hairpins, which interact via knobs-into-holes packing between helices Ai and B(i-1), as is characteristic of TPR hairpins. A superposition to the CTPR3

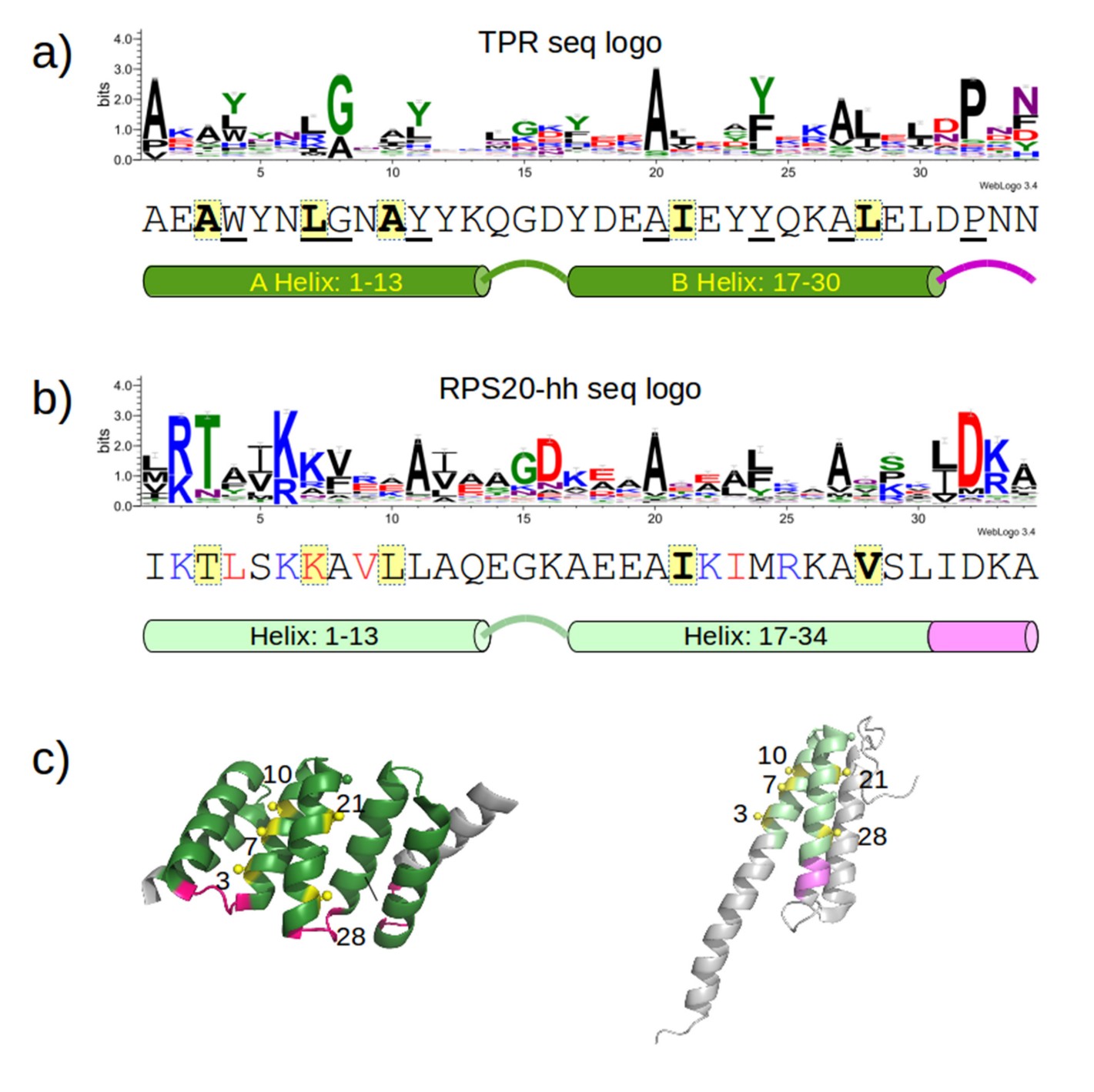

**Figure 4.** Sequence positions considered for optimizing the designed proteins. (a) Sequence logo of the TPR motif. A TPR consensus sequence (*Main et al., 2003b*) (PDB: 1na0, chain A) and its secondary structure determined by DSSP (*Kabsch and Sander, 1983*) are aligned below the sequence logo. The eight TPR signature positions are underscored in the consensus sequence. The five interface positions are highlighted in yellow. (b) Sequence logo of RPS20-hh. The RPS20-hhta sequence and its predicted secondary structure using Quick2D (*Biegert et al., 2006*) is aligned below the sequence logo. The derived interface positions are highlighted in yellow. The four residues subjected to mutations are colored in red. The four positively charged residues selected for mutation to lower the surface charge are in blue. (c) The locations of the interface positions displayed on a TPR (left) and a RPS20 structure (right). In both structures, the interface positions are labeled and highlighted as yellow spheres. The TPR structure is CTPR3 (PDB: 1na0, chain A), which is shown as a cartoon and is colored using the same scheme as the secondary structure representation in (a). The stop helix is in gray. The RPS20 structure is from *T. thermophilus* (PDB: 4gkj, chain T), in which the RPS20-hh fragment is colored using the same scheme as the secondary structure representation in (b). The sequence logos were generated using WebLogo (*Crooks et al., 2004*). Sequences from representative proteome

*Figure 4 continued on next page*

*Figure 4 continued*

75% (*Chen et al., 2011*) downloaded from Pfam families *TPR_1* and *Ribosomal_S20p* were used as input to WebLogo (9338 and 972 sequences, respectively). The structures were rendered using PyMOL (*Schrödinger, 2010*).

The following figure supplements are available for figure 4:

**Figure supplement 1.** Mutual information plot (**a** and **b**) and direct coupling analysis plot (**c** and **d**) for TPR repeat sequences.

**Figure supplement 2.** Rosetta energy scores (*fixbb+relax*) for TPR designs based on RPS20-hhta sequence and various sets of mutations.

**Figure supplement 3.** Prediction of intrinsically disordered regions in RPS20 of *Thermus aquaticus* (NCBI gi: 489134531, accession: WP_003044315.1) using **a**) IUPred (http://iupred.enzim.hu/); **b**) DisEMBL (http://dis.embl.de/) and **c**) PONDR (http://www.pondr.com/).

structure yields $C_\alpha$ RMSD values below 2.6 Å (*supplementary file 1I*). An unexpected difference to the canonical TPR structure is that the stop helix of M4N is not resolved in any of the three chains. However, this missing helix is compensated for by a specific dimerization mode of two M4N protomers. Therein, the C-terminal TPR units of the two protomers form a tight interface, in which the B3 helix of each chain substitutes for the stop helix of the other, mimicking the capping effect of the stop helix (*Figure 6*). A superposition of this mimicry to the last TPR unit and stop helix of CTPR3 yields $C_\alpha$ RMSD values as low as 1.2 Å over 44 residues. The third chain of the ASU, however, was found as a monomer, capping its C-terminal TPR unit in a more unspecific manner by packing it orthogonally against the two A1 helices of the dimer (*Figure 5a*).

Analysis by mass spectrometry revealed that the M4N stop helix had been partially proteolyzed upon expression of the protein (*Figure 5—figure supplement 3*). Although we did not observe proteolysis in the other folded constructs (M2, M4E and M5), which were also all dimeric, we analyzed whether proteolysis might have favored the dimerization of M4N. Extending the stop helix with a C-terminal His$_6$-tag prevented proteolysis, but did not affect stability or dimerization (M4N-His; *Supplementary file 1F*). We conclude that in the amplified constructs, the observed interactions are more favorable than the interaction with the native stop helix, releasing it and rendering it prone to degradation. This led us to ask whether this helix is in fact dispensable. Indeed, an M4NΔC construct, which terminates with the B3 helix, showed the same stability and dimerization as M4N. We obtained two structures for M4NΔC from different crystal forms at 2.0 Å and 1.7 Å resolution, respectively, the first (CF I) with two dimers in the ASU and the second (CF II) with a single chain in the ASU, for which we constructed the dimer by crystallographic symmetry. All three dimers superimpose to the M4N dimer with $C_\alpha$ RMSD below 1.9 Å (*Figure 7*, *Supplementary file 1I*). We conclude that the stop helix is dispensable for folding, dimerization and the stability of our designed constructs.

The geometry of dimerization in M4N has not been observed in TPR structures before. Although there have been reports on the self-association of TPR-containing proteins involved in various regulatory biological processes (*Bansal et al., 2009a*, *2009b*; *Ramarao et al., 2001*; *Serasinghe and Yoon, 2008*), only a small number of oligomeric TPR structures have been determined to date (*Krachler et al., 2010*; *Lunelli et al., 2009*; *Zeytuni et al., 2012*, *2015*; *Zhang et al., 2010*). None of these resemble the ring-shaped dimer of M4N.

## Mutations introduced into RPS20-hhta are neutral to Thermus

The results shown above suggest that the mutations we made to RPS20-hhta were crucial for obtaining the TPR fold. If RPS20 and TPR proteins indeed share a common ancestor, such mutations may have been sampled in the course of evolution. Since we cannot reconstruct the ancestor and do not know what its function was beyond a general expectation of RNA binding, we decided to test whether the mutations we introduced impaired the interaction between RPS20 and its cognate RNA, as an indication of their compatibility with RNA interaction. Each mutation in M2 and M4N occurs in natural RPS20 sequences (see *Supplementary file 1A*), but no RPS20 sequence has all four mutations simultaneously and we therefore tested if they can be tolerated in vivo. As genetic engineering in *T. aquaticus* turned out to be unfeasible, we performed these tests in *T. thermophilus* HB8, which

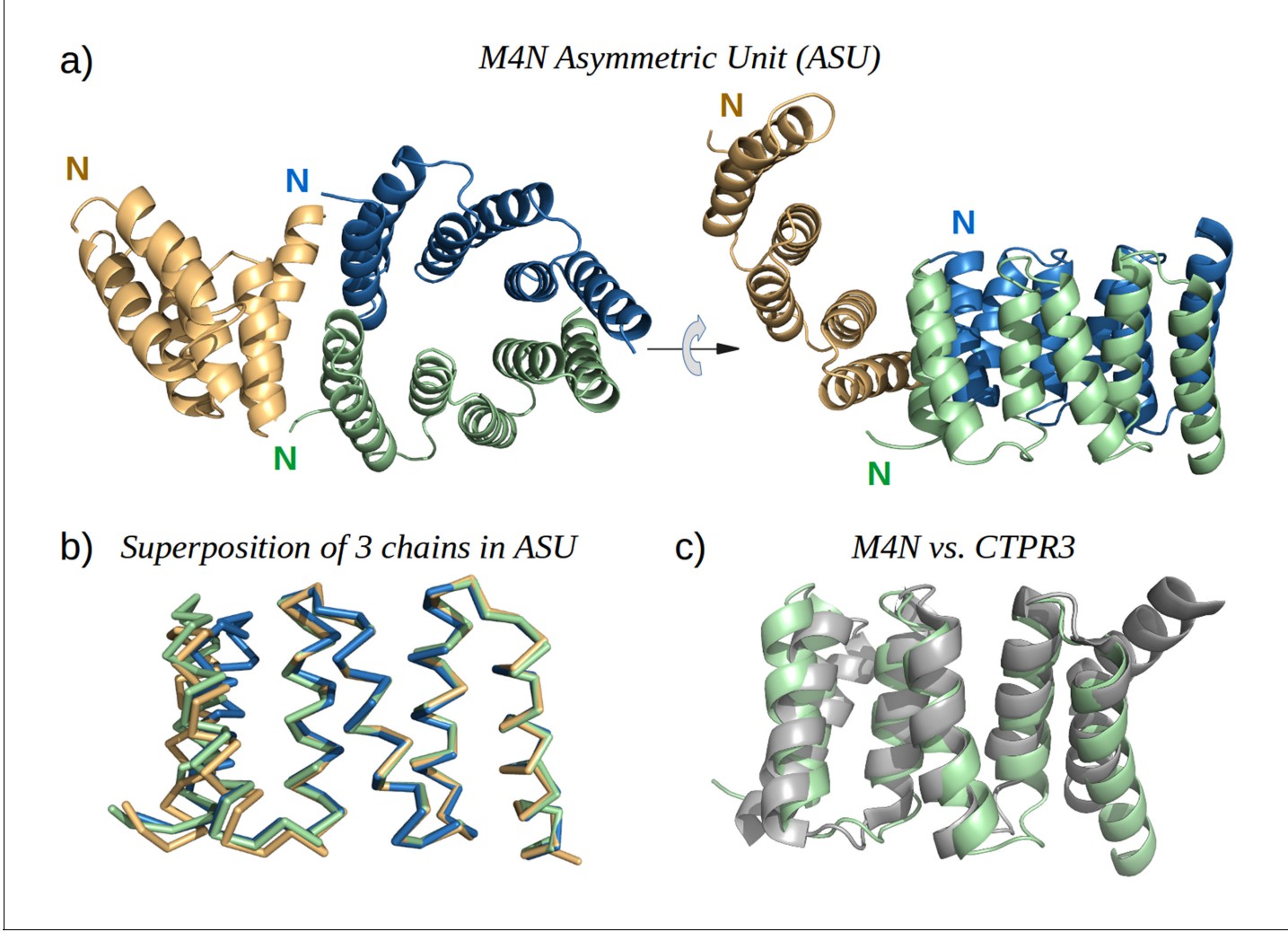

**Figure 5.** The X-ray structure of M4N. (a) The three chains A, B and C in the asymmetric unit are colored green, blue and yellow, respectively. Chains A and B form a dimer. (b) Superposition of the three chains. Only Cα traces are shown for clarity. (c) Superposition of M4N (chain A, green) and the designed consensus TPR CTPR3 (PDB: 1na0, chain A, gray).

The following figure supplements are available for figure 5:

**Figure supplement 1.** The interaction of M4N molecules in the crystal.

**Figure supplement 2.** Urea denaturation of designed TPR repeats.

**Figure supplement 3.** Mass spectrometry (MS) analysis of M4N.

is a well-established model organism. The RPS20 helical hairpins in *T. aquaticus* and *T. thermophilus* differ only at four positions, of which two are highly conservative substitutions (*Figure 8a*).

We first attempted to substitute the chromosomal RPS20-encoding gene, *rpsT*, with a kanamycin resistance cassette, to obtain *T. thermophilus* strain KM4 (*Figure 8b*). For complementation we introduced plasmids bearing wild type *rpsT* from *T. thermophilus* (TT) or *T. aquaticus* (TA), *rpsT* from *T. aquaticus* carrying the mutations from M2 (TA2) or M4N (TA4), or merely empty plasmids as negative control (E). We monitored the substitution of *rpsT* by a PCR screening protocol, which will amplify a 1500 bp region if WT *rpsT* is substituted and an 800 bp region otherwise (*Figure 8b*). Under selection pressure from kanamycin, only the 1500 bp product was obtained in all cases where

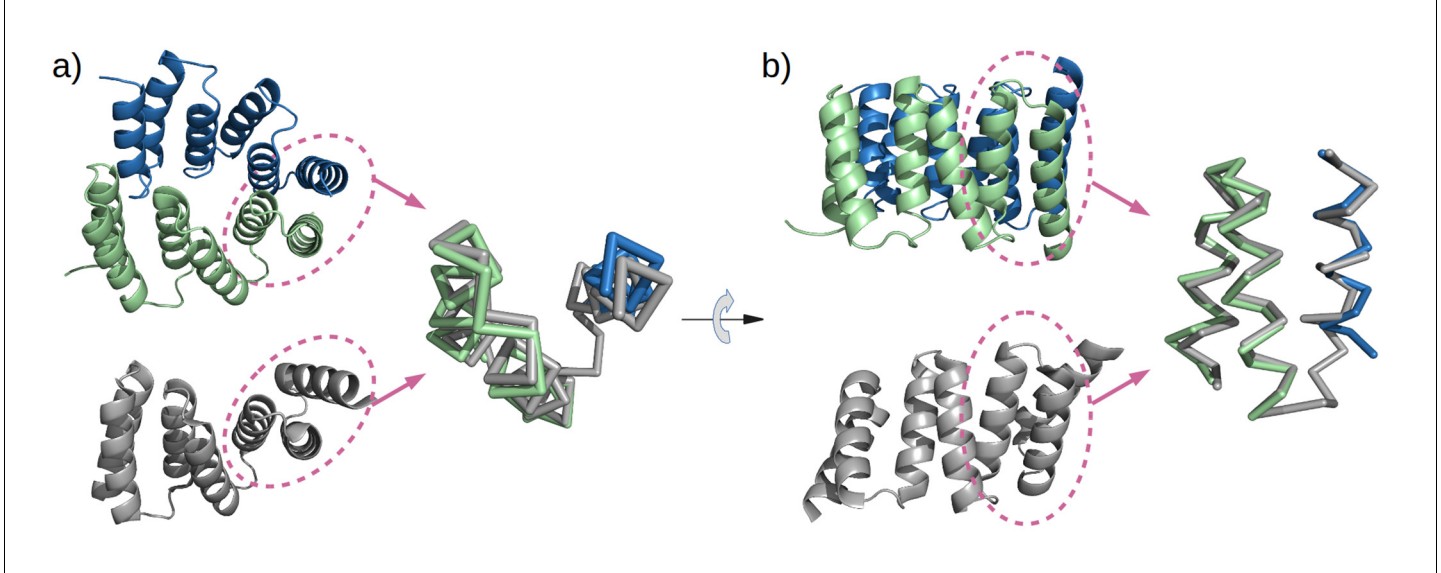

**Figure 6.** Mimicry of the stop helix in the M4N dimer. The C-terminal TPR unit in chain A (green) and the C-terminal helix B3 in chain B (blue) are superposed to the last TPR unit plus the stop helix in CTPR3 (gray).

plasmid-borne *rpsT* was introduced, whether in wild-type or mutated form (*Figure 8c* panels 1 and 2, lanes TT, TA, TA2 and TA4), showing that the chromosomal gene had been fully substituted. In contrast, PCR screening of strain KM4 complemented with an empty plasmid produced both 800 bp and 1500 bp fragments (*Figure 8c* panels 1 and 2, lane E). Since *T. thermophilus* HB8 is a polyploid organism (minimally tetraploid [*Ohtani et al., 2010*]), this result shows that *rpsT* can be reduced in copy number, but not fully eliminated, suggesting that the gene is essential.

To assess the level of substitution achieved with the various plasmids, we designed a second PCR screening protocol to specifically detect chromosomal *rpsT* via a 300 bp product. At low kanamycin concentrations this protocol always generated a product (*Figure 8d* panel 1), but at increased kanamycin concentration we did not obtain product for any *rpsT* allele (*Figure 8d* panel 2, lanes TT, TA, TA2 and TA4). This demonstrates that plasmid-borne *rpsT* and its mutants were able to complement the chromosomal *rpsT* and that the latter was displaced from the population to a level that left it undetectable by PCR. In contrast, we could never completely suppress chromosomal *rpsT* in strain KM4 complemented with an empty plasmid, even under high kanamycin conditions (120 μg/ml).

In *E. coli* and *Salmonella enterica*, *rpsT* has been reported to be non-essential, but its deletion significantly lowers growth rates (*Bubunenko et al., 2007*; *Tobin et al., 2010*). We found that *rpsT* is essential in *T. thermophilus*, but that its loss could be complemented by wild-type and mutant *T. aquaticus rpsT*, and that this complementation restored wild-type levels of growth (*Figure 8e*). Moreover, when the selection pressure from kanamycin was removed, no reversal in the PCR products was detected for any strain (*Figure 8c and d*, panel 3), which confirms that chromosomal *rpsT* was substantially displaced during kanamycin treatment. We conclude that *rpsT* from *T. aquaticus* and its two mutated alleles are neutral with respect to survival and growth for *T. thermophilus*. This demonstrates that the mutations we introduced do not affect negatively the interaction between RPS20 and its cognate RNA, and that therefore such mutations could have been sampled multiply and in a cumulative fashion by neutral drift during the course of evolution.

## Implications for the emergence of folded proteins

Proteins are the most complex macromolecules synthesized in nature and by and large need to assume defined structures for their activity. This folding process is complicated and easily disrupted, witness the elaborate systems for protein folding, quality control and degradation universal to all living beings. Despite the widespread problems to reach and maintain the folded state, natural proteins nevertheless form a best-case group, since the overwhelming majority of random polypeptides

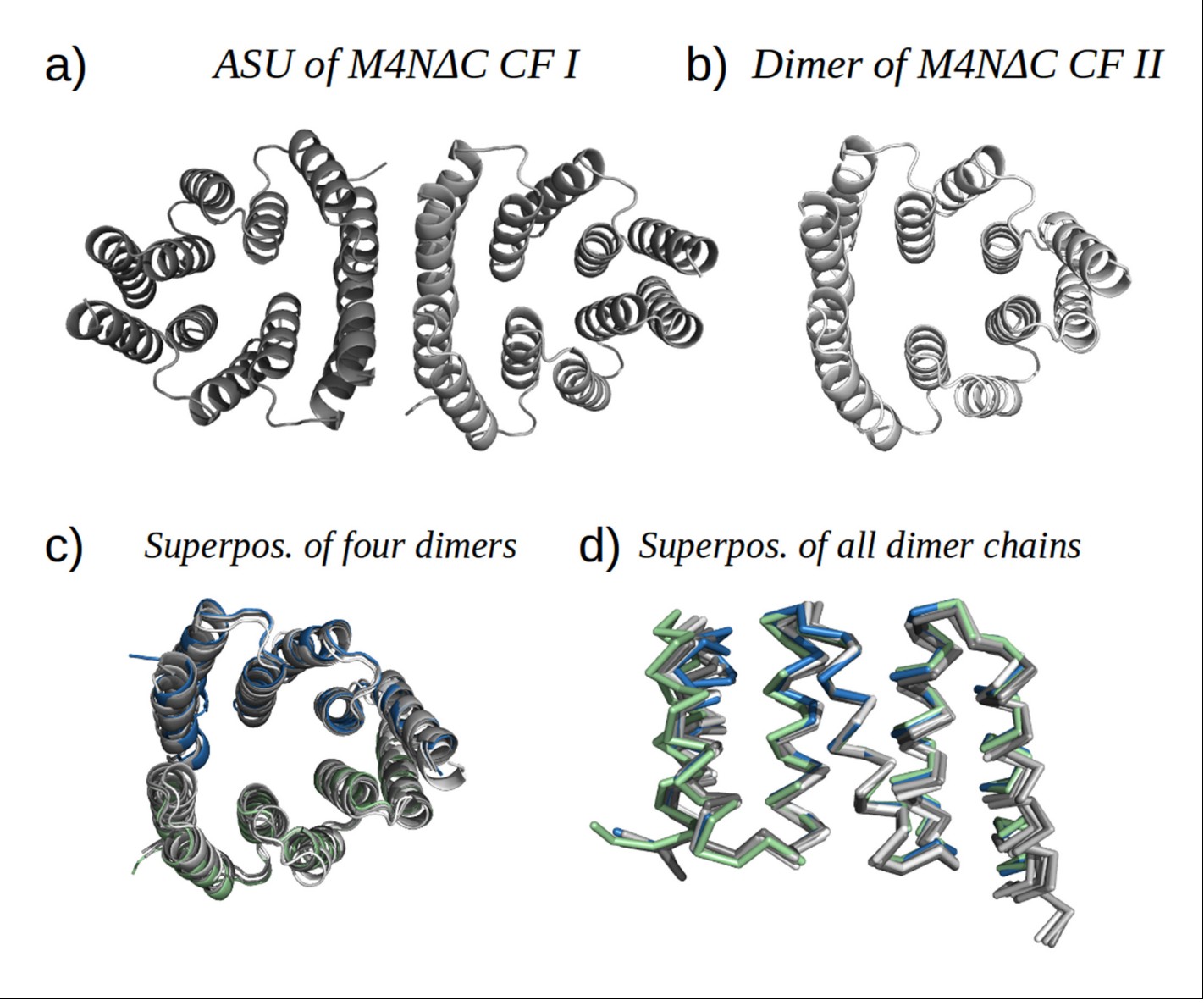

**Figure 7.** M4NΔC structures of two different crystal forms and their comparison to the M4N dimer. (**a**) Two dimers in the ASU of M4NΔC CF I. (**b**) Dimer constructed by applying the crystallographic symmetry to the single chain in the ASU of M4NΔC CF II. (**c**) Superposition of all the four M4N and M4NΔC dimers. The M4N dimer is in green and blue. The three M4NΔC dimers are in different shades of gray as in (**a**) and (**b**). (**d**) Superposition of all the chains in the M4N and M4NΔC dimers (eight chains in total). Only Cα traces of proteins are shown for clarity.

do not appear to have a folded structure (*Keefe and Szostak, 2001*; *Wei et al., 2003*). It thus seems impossible that, at the origin of life, the prototypes for the folded proteins we see today could have arisen by random concatenation of amino acids. We have proposed that folding resulted from the increasing complexity of peptides that supported RNA replication and catalysis, and that these peptides assumed their structure through the interaction with the RNA scaffold (*Lupas et al., 2001*; *Söding and Lupas, 2003*). In this view, protein folding was an emergent property of RNA-peptide coevolution. We have recently described 40 such peptides whose conservation in diverse folds suggests that they predated folded proteins (*Alva et al., 2015*). These peptides are enriched for nucleic-acid binders, particularly in the context of the ribosome.

Due to its extremely slow rate of change, the ribosome essentially represents a living fossil, providing the possibility to study the chronology of ancient events in molecular evolution (*Hsiao et al.,*

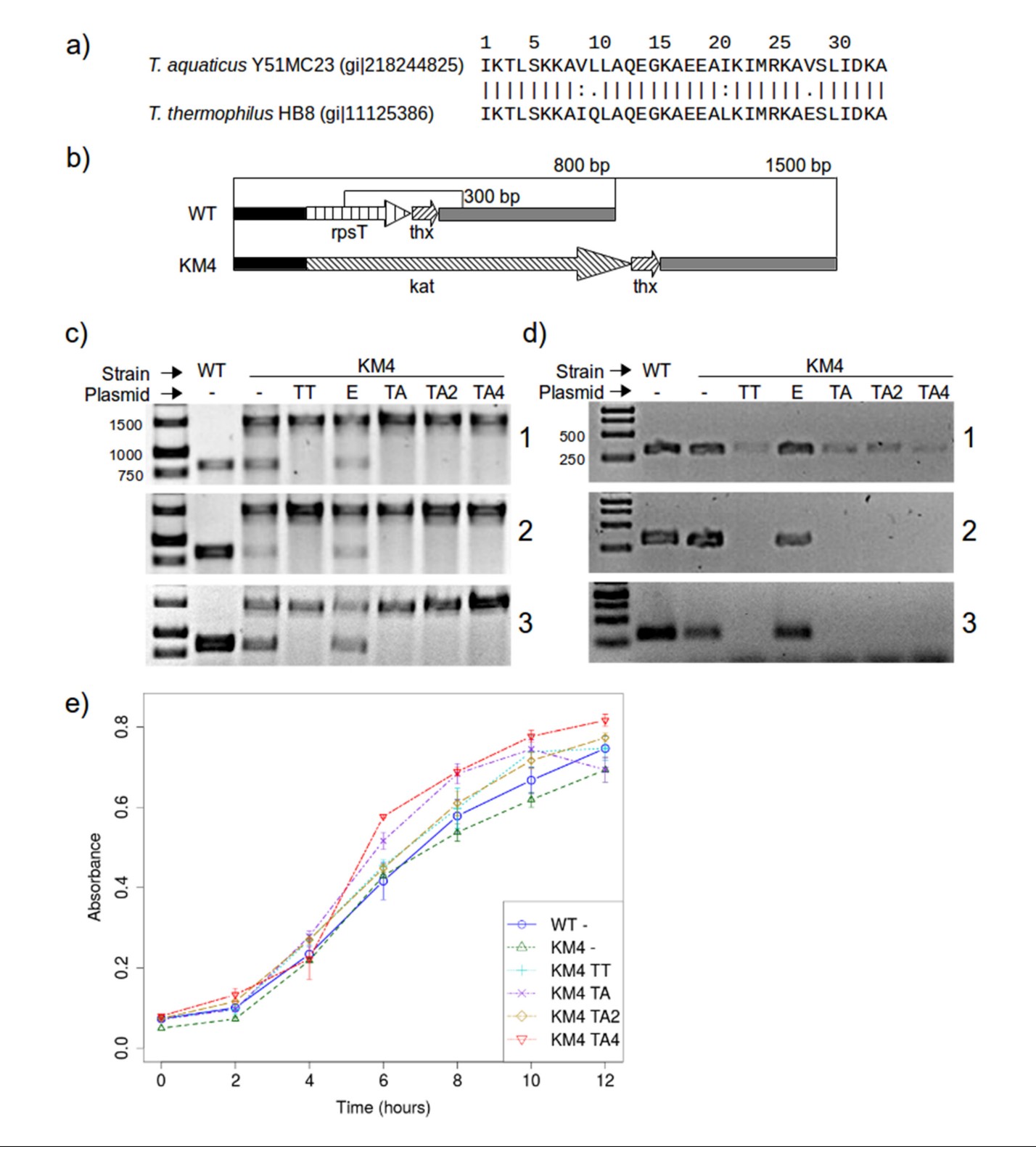

**Figure 8.** RPS20 variants M2 and M4N are functional proteins. (**a**) The 34 amino-acid long RPS20-hh fragments in *T. aquaticus* and *T. thermophilus* differ only at four positions, including two conservative mutations (V9I and I21L). (**b**) Scheme of the *rpsT* region before (upper) and after (lower) substitution of *rpsT* with the kanamycin resistance cassette (*kat*). Base pair (bp) values indicate the PCR products that can be amplified. Regions depicted with the same pattern are identical. Regions in solid black and gray also contain genes which are not marked for clarity. (**c**) PCR to detect substitution of rps20 by the *kat* gene and (**d**) PCR to detect the presence of chromosomal *rpsT* in *T. thermophilus* strains (WT: *T. thermophilus* HB8; KM4:*T. thermophilus*

*Figure 8 continued on next page*

*Figure 8 continued*

KM4) carrying various plasmids (TT: pJJSpro-rpsTTt; E: pJJSpro; TA: pJJSpro-rpsTTa; TA2: pJJSpro-rpsTTaM2; TA4: pJJSpro-rpsTTaM4N; -: No plasmid) after sequential grow under different selective pressures (1: 30 µg/ml kanamycin; 2: 120 µg/ml kanamycin; 3: 0 µg/ml kanamycin). (**e**) Corresponding growth curves of the host bacteria with various substitutions and plasmids.

*2009*). Thus, core ribosomal proteins offer a window into the time when proteins were acquiring the ability to fold. Those close to the catalytic center almost entirely lack secondary structure. Further away from the center, their secondary structure content gradually increases and at the periphery, these secondary structure elements become arranged into topologies that parallel those seen in cytosolic proteins (*Hsiao et al., 2009*). Collectively, the structures of ribosomal proteins chart a path of progressive emancipation from the RNA scaffold. Even the peripheral proteins, however, still mostly assume their structure only in the context of the ribosomal RNA, as exemplified by RPS20 in our study (*Supplementary file 1F*, see also *Paterakis et al., 1983*).

The simplest mechanism to achieve an increase in complexity is the repetition of building blocks and nature provides many examples for this, at all levels of organization. The dominant role of repetition in the genesis of protein folds has been documented in many publications since the 1960s (*Alva et al., 2007*; *Blundell et al., 1979*; *Broom et al., 2012*; *Eck and Dayhoff, 1966*; *Kopec and Lupas, 2013*; *Lee and Blaber, 2011*; *McLachlan, 1972*, *1987*; *Remmert et al., 2010*; *Söding et al., 2006*). As a test of this mechanism, we explored whether a peptide originating from a ribosomal protein that is disordered outside the context of the ribosome, could form a folded protein through an increase in complexity afforded by repetition. For this, we chose a present-day representative of one of the 40 fragments we reconstructed (*Alva et al., 2015*); this fragment is naturally found in a single copy in several different folds, including that of ribosomal protein RPS20, and repetitively in one fold, TPR. Simple repetition was not sufficient in our case, but the repeat protein was so close to a folded structure that only two point mutations per repeat were necessary to allow it to fold reliably. The mutations needed for this transition did not appear to affect negatively the interaction with the RNA scaffold, raising the possibility that they could have been among the variants sampled multiply in the course of evolution.

Our experiments recapitulate a scenario for the emergence of a protein fold by a widespread and well-documented mechanism, and show that this could have proceeded in a straightforward way. These experiments represent a proof of concept, starting with a modern peptide likely to still retain many features of an ancestral αα-hairpin that gave rise to a number of folds, including TPR. Rather than proposing proto-RPS20 as the parent of TPR domains, we see it as one of many proteins emerging from this ancestral hairpin. Given the ease with which repetition of the RPS20 hairpin yielded a TPR-like fold, we consider it likely that the hairpins belonging to the ancestral group were amplified many times during the emergence of folded proteins to yield a range of TPR-like offspring, of which only one may have survived to this day (but see also the figure legend to *Figure 1*). The reason for this limited survival may lie in the fact that a structure is a prerequisite for protein function, but it is the function that is under biological selection. It could be that the newly emerged TPR-like folds required many additional changes to achieve a useful activity and that therefore only very few – possibly just one – survived. We consider a different scenario more probable, however. All present-day TPR domains whose function has been characterized mediate protein-protein interactions by binding to linear sequence motifs in unstructured polypeptide segments (*D'Andrea and Regan, 2003*; *Zeytuni and Zarivach, 2012*). This activity would have been particularly relevant at a time of transition from peptides dependent on RNA scaffolds for their structure, to autonomously folded polypeptides. Functions relevant in this context would have been to prevent aggregation and increase the solubility of newly emerging (poly)peptides, to promote autonomous folding, to serve as assembling factors for RNA-protein and protein-protein complexes, and to recognize targeting sequences in the emerging cellular networks. It therefore seems likely to us that many of the newly evolved TPR-like folds became established in one or the other of these activities, only to be subsequently displaced by folding becoming a general property of cellular polypeptides and by more advanced, energy-dependent folding factors, which offered much better regulation. Exploring the extent to which our new TPR protein could fulfill such functions represents the next frontier in our studies.

# Materials and methods

## Phylogeny for recently amplified TPR arrays

All sequence similarity searches in this work were performed using the Web BLAST (RRID:SCR_004870) from the National Institute for Biotechnology Information (NCBI; http://blast.ncbi.nlm.nih.gov; *Boratyn et al., 2013*) and in the MPI Bioinformatics Toolkit (RRID:SCR_010277, https://toolkit.tuebingen.mpg.de/; *Alva et al., 2016*). Examples of recently amplified repeat units in TPR were taken from a previous investigation (*Dunin-Horkawicz et al., 2014*). The TPR domain in serine/threonine-protein phosphatase 5 was chosen as a representative three-repeat TPR, the most common TPR form in natural proteins (*D'Andrea and Regan, 2003*; *Sawyer et al., 2013*), to study divergent evolution of TPR. We ran BLAST on the non-redundant protein sequence database (nr) with an E-value threshold of 0.05 using the TPR domain of serine/threonine-protein phosphatase 5 from *Homo sapiens* as query (*Das et al., 1998*). From the results, we chose seven taxa to cover a diverse range of life.

TPRpred program (*Karpenahalli et al., 2007*) was used to help identify tandem repeats of TPR units. The construction of multiple sequence alignments (MSAs) for TPR units was straightforward as all TPR units are of the same size (34 aa) and no indels were allowed in the MSAs. We used Clustal X 2.1 (*Larkin et al., 2007*) to build phylogenetic trees using the neighbor-joining clustering algorithm and 1000 bootstrap trials (Bootstrap N-J Tree). SplitsTree4 (*Huson and Bryant, 2006*) was used to render the phylogenetic trees.

## Identification of helical hairpins resembling the TPR unit

To find proteins homologous to the TPR unit, we first employed the TPRpred program (*Karpenahalli et al., 2007*) to identify proteins that share significant sequence similarity to the TPR sequence profile, then filtered them by comparing to the TPR structures.

First, TPRpred program with TPR profile tpr2.8 was used to identify TPR unit like sequences from all protein sequences of known structures in the Protein Data Bank (PDB, RRID:SCR_012820) (*Berman et al., 2000*). Protein sequences from the SEQRES record in PDB files were downloaded from the PDB. We only considered sequences with at least 34 residues, which is the length of the TPR unit. Redundancy was removed by keeping only non-identical sequences. In total, 68,197 sequences were scanned by using TPRpred with default parameters. Only fragments predicted to be TPR with a p-value lower than $1.0e-4$ were retained (646 hits). We estimated the false discovery rate (FDR) (*Noble, 2009*) associated with this p-value cutoff using a simulated sequence dataset generated by using the amino-acid composition derived from the PDB sequences. The dataset contains the same number of sequences of the same length distribution as the PDB sequences. The FDR was estimated to be the ratio of the number of hits in the simulated dataset to the number of detected hits in the PDB sequences (*Noble, 2009*). We repeated the simulation 100 times and the FDR was estimated to be $1.0 \pm 0.4\%$.

Within the 646 hits, we kept only TPR unit singletons, which are TPR units that have no other TPR units close to them within a distance of 10 residues in the same sequence. TPR units of identical sequences are considered only once. Subsequently, these TPR unit singletons were filtered by removing those annotated to belong to clan *TPR* (CL0020) in Pfam 27.0 (RRID:SCR_004726).

The structures of the predicted TPR units obtained from the previous step were then compared to an average TPR unit structure. A predicted TPR unit was discarded if the $C_\alpha$ RMSD of the 34 residues is greater than 2.0 Å after superposition. The average TPR unit structure was generated by considering all proteins belonging to family tetratricopeptide repeat (TPR) (a.118.8.1) in SCOP 1.75 (RRID:SCR_007039) (*Murzin et al., 1995*). TPR repeats in these proteins were again detected using TPRpred and a per-repeat p-value cutoff of $1.0e-4$ was used. In total, 50 non-redundant TPR repeat fragments were identified and superposed using a multiple structure alignment tool MultiProt (*Shatsky et al., 2004*). The average $C_\alpha$ positions were calculated from the 50 structures after superposition. We obtained 31 fragments after the structure filtering step (*Supplementary file 1C*). We then inspected the protein structures using PyMOL (RRID:SCR_000305) (*Schrödinger, 2010*). Among them, 22 were observed to be involved in the formation of solenoid or tandem repeat structures and were thus not further considered.

## Identification of TPR homologs in RPS20

We applied TPRpred to scan all RPS20 sequences belonging to Pfam 27.0 family *Ribosomal S20p* (PF01649), including sequences from both datasets 'full' and 'ncbi'. There are 4402 and 2284 sequences in the two sets. We merged the two sets and removed identical sequences to create a dataset of 3742 RPS20 sequences. TPRpred was used to detect TPR unit homologs in them. We obtained 24 hits in these RPS20 sequences predicted by TPRpred to match TPR unit profile with a p-value smaller than 1.0e−4 (see *Supplementary file 1D*).

We defined 'interface positions' in the TPR unit and then transferred the definition to RPS20-hh according to their structure superposition. We considered the residues on the outer side of the two helices facing neighboring TPR units. Both helix A and helix B in the TPR unit are α-helices, which have on average 3.6 residues per turn. Thus, every third or fourth residue always appears on the same side of the helix. They are positions 3, 7 and 10 in helix A and positions 17, 21, 24 and 28 in helix B. According to the TPR sequence profile compiled by Main et al. (*Main et al., 2003b*), the most common residues at these positions are hydrophobic except for positions 17 and 24, where the most common residues are both Tyr (see also *Figure 4a*). Therefore, positions 17 and 24 were not included in the definition of interface positions. Furthermore, the residue at position equivalent to position 24 in RPS20 structure faces its C-terminal helix and is already an interface residue (*Figure 4c*). Thus, it was not considered as an interface position to be checked in the study. In the end, only positions 3, 7, 10, 21 and 28 in RPS20-hh were defined to be interface positions to be examined, because they are exposed to the solvent or interact with the RNA molecules in the ribosome, but would interact with neighboring repeats in the TPR fold.

We searched all RPS20 sequences in Pfam 27.0 family *Ribosomal_S20p* (PF01649), including both datasets 'full' and 'ncbi', for candidates in which the interface positions are occupied by as many hydrophobic residues as possible. In the MSA provided by Pfam, we extracted the 34 columns that correspond to the sequence fragment of RPS20-hh from *Thermus aquaticus*, which was found by TPRpred to be the hit with the best p-value and was thus used as the reference RPS20-hh. We obtained 1370 sequence fragments that do not contain any indels, in which the interface positions were examined for hydrophobicity. Here, Ala, Ile, Leu, Met, Phe, Val were considered as hydrophobic residues. Trp was not included as its side chain may be too large to be accommodated at the interface.

We employed several low-complexity / intrinsically disordered region prediction methods (SEG [*Wootton, 1994*], PONDR [*Romero et al., 2001*], DisEMBL [*Linding et al., 2003*], IUPred [*Dosztányi et al., 2005a*, *2005b*]) to investigate putative intrinsically disordered regions in the RPS20 of *Thermus aquaticus*. We ran SEG with three sets of recommended parameters (*Wootton and Federhen, 1996*) and the other approaches with default parameters.

## Optimization of RPS20-hh in the designed TPRs

We considered eight positions (2, 4, 6, 7, 9, 22, 23 and 25) in RPS20-hhta for optimization apart from the four residues at the C-terminus.

Main et al. (*Main et al., 2003b*) discovered a set of eight 'TPR signature residues' in the consensus design: W4, L7, G8, Y11, A20, Y24, A27 and P32. Six of them are missing in RPS20-hhta except A20 and A27. Following the principle of consensus design, we introduced L4W and K7L into RPS20-hhta. K7 is also one of the interface positions that ought to be mutated to hydrophobic residue for better packing at interfaces. A8 and L11 were not optimized because they are the second and third most common residues at positions 8 and 11 in the TPR profile, respectively. M24 was also retained because it seems long hydrophobic side chains are favored at position 24 though Met is not one of the three most common residues (YFL). P32 was introduced to replace D32 in RPS20-hhta as part of the C-terminal consensus loop (DPNN) between repeats.

Co-evolution is commonly observed between physically interacting residues (*de Juan et al., 2013*). We investigated if any positions we optimized are involved in a co-evolution relationship so that we can preserve such correlations. We performed a direct coupling analysis (*Morcos et al., 2011*) and computed the mutual information using MatrixPlot (*Gorodkin et al., 1999*) between all positions in TPR repeat sequences. The results of both approaches revealed that the highest correlation occurs between positions 7 and 23 (*Figure 4—figure supplement 1*), with the most commonly observed combinations being R7-D23 and L7-Y23. Therefore, we always mutated I23 to the most

commonly observed residue tyrosine (I23Y) in the TPR consensus sequence together with aforementioned mutation K7L. In addition, we considered combination K7R and I23D together. Combination K7-I23D was also tested because of highly similar physicochemical properties between Lys and Arg side chains.

The hydrophobic side chain of valine at position 9 in RPS20-hhta is buried between helices in RPS20, but would be exposed on the surface of the designed protein except in the last repeat, in which V9 interacts with the stop helix. Therefore, it is considered to be mutated to the most common residue asparagine (V9N) in the TPR repeat consensus except in the last repeat (*Figure 4c*).

RPS20-hhta sequence and surface is enriched with positively charged residues (*Figure 4b*). This would lead to the exceedingly high theoretical iso-electric point (pI) of the designed proteins. Natural TPR proteins tend to exhibit zero net charge (*Magliery and Regan, 2004*). Hence, we decided to randomly mutate the positively charged residues (Lys and Arg) in the two helices of RPS20-hhta to the corresponding most common residues in TPR sequence profile (K2E, K6N, K22E, R25Q/E). K26 was not mutated as Lys is already the most common residue in the TPR profile.

At the C-terminus of the designed TPR, the last four residue of RPS20-hhta (IDKA) were replaced with the TPR consensus loop sequence (DPNN) between repeat units. The reason is as follows. The secondary structure of the TPR unit is helix (13 aa) – loop (3aa) – helix (14 aa) – loop (4aa), while the secondary structure of the RPS20-hhta identified to be homologous to TPR unit is helix (13 aa) – loop (3 aa) – helix (18 aa) (*Figures 2* and *4*). The last four residues may have been included in the prediction by TPRpred merely to fulfill the size requirement of TPR repeat (34 aa). Indeed, when we scanned RPS20-hhta sequence using the hidden Markov model constructed for Pfam family *TPR_1*, only positions 2–28 were found to be similar to the *TPR_1* profile using HMMER 3.0 (RRID:SCR_005305) (*Eddy, 2009*), even if all filters were switched off. So the four very C-terminal residues in RPS20-hhta were not used in the designed TPR between repeat units. They were not replaced in the last repeat unit (*Figure 3*).

## Structure modeling and refinement in silico

CTPR3 structure of an idealized TPR repeat (*Main et al., 2003b*) (PDB id: 1na0, chain A) was taken as the main template to build an initial TPR structure model using RPS20-hhta. Helix B3 and the stop helix in our designed protein are different from natural TPRs, but more similar to natural RPS20s. So we also used a RPS20 protein as the structure template for the last repeat and the stop helix. The structure of RPS20 from *Thermus thermophilus* HB8 (PDB id: 2vqe, chain T) was used because it was the structure with the best resolution (2.5 Å). The C-terminal loop in 2vqeT was discarded. The two structures 1na0A and 2vqeT were merged into a hybrid template based on the superposition of their homologous helical hairpins: the third TPR unit in 1na0A and the RPS20-hh in 2vqeT (the very C-terminal four residues were not used). We then modeled the designed TPR sequences using RPS20-hhta onto the hybrid structure template using Rosetta programs *fixbb* and *relax* (*Das and Baker, 2008*). The Rosetta fixed backbone design application *fixbb* was used to make the initial model. Subsequently, these models were relaxed using the Rosetta structure refinement application *relax*. The two steps were iterated three times. See the *Supplementary file 1E* for the command lines. Rosetta 3.4 was used in the work.

We selected five constructs for further testing in vitro (*Table 1*). They are among the best-scoring constructs according to the in silico simulation (*Figure 4—figure supplement 2*). If two constructs have comparable scores (they are adjacent in the score ranking), the one with fewer mutations was preferred. The selected constructs all differ at least at two positions in their sequences. When we searched these optimized RPS20-hhta fragments in the NCBI *nr* database using BLAST (*Camacho et al., 2009*), the top hits were still RPS20s.

## Cloning, protein expression and purification

DNA sequences coding for the designed TPR repeats were gene-synthesized in codon-optimized form (Eurofins) and cloned into vector pET-28b (Novagen) using NcoI/HindIII restriction sites, and into pETHis_1a to generate proteins with an N-terminal cleavable His$_6$-tag. RPS20 *T. aquaticus* and *T. thermophilus* genes were amplified from genomic DNA and cloned likewise. Recombinant plasmids were transformed into *E. coli* strain BL21-Gold (DE3) grown on LB agar plates containing 50

μg/ml kanamycin. For expression, cells were cultured at 25°C and induced with 1 mM isopropyl-D-thiogalactopyranoside (IPTG) at an $OD_{600}$ of 0.6 for continued growth overnight.

Bacterial cell pellets were resuspended in buffer A (50 mM Tris pH 8, 150 mM NaCl), supplemented with 5 mM $MgCl_2$, DNaseI (Applichem) and protease inhibitor cocktail (cOmplete, Roche). After breaking the cells in a French Press, the suspension was centrifuged twice at 37,000 g. Soluble $His_6$-tagged proteins were purified by binding proteins to Ni-NTA columns (GE Healthcare) in buffer A (50 mM Tris pH 8.0, 300 mM NaCl) and elution with increasing concentrations of imidazole up to 0.6 M. Eluted proteins were dialyzed against buffer A for cleavage by $His_6$-TEV-protease (50 U/mg protein). Cleavage leaves two additional residues (Gly-Ala) as N-terminal extension to all proteins produced in this manner. After incubation overnight, cleaved proteins were re-run on Ni-NTA columns and collected in the flow-through. They were finally purified by gel size exclusion chromatography (Superdex G75, GE Healthcare) in buffer A containing 0.5 mM EDTA. Insoluble proteins were dissolved in 6 M guanidinium chloride and refolded by dialysis overnight against buffer A. Refolded proteins were further purified by sequential anion-exchange (Q Sepharose HP) and cation-exchange (SP Sepharose HP) chromatography using 0–500 mM NaCl salt gradients in buffer D (20 mM Tris pH 8, 1 mM EDTA), and by gel size exclusion chromatography (Superdex G75) in buffer A.

## Biophysical characterization

To determine the native molecular mass of designed TPR repeats, static light scattering experiment was performed by applying samples onto a superdex S200 gel size exclusion column to which a mini-DAWN Tristar Laser photometer (Wyatt) and an RI 2031 differential refractometer (JASCO) were coupled. Runs were performed in buffer A. Data analysis and molecular mass calculations were carried out with ASTRA V software (Wyatt). Tryptophan fluorescence spectra were recorded on a Jasco FP-6500 spectrofluorometer at 23°C; excitation was at 280 nm, emission spectra were collected from 300–400 nm. Circular dichroism (CD) spectra from 200–250 nm were recorded with a Jasco J-810 spectropolarimeter at 23°C in buffer E (30 mM MOPS pH 7.2, 150 mM NaCl). Cuvettes of 1 mm path length were used in all measurements. For melting curves and determination of Tm, CD measurements were recorded at 222 nm from 20–95°C, the temperature change was set to 1°C per minute, using a Peltier-controlled sample holder unit. For equilibrium-unfolding experiments performed at 23°C, native protein was mixed with different concentrations of urea in buffer A. After equilibration, circular dichroism was monitored at 222 nm. The fraction of unfolded protein $f_U$ was determined based on $f_u = (y_F - y)/(y_F - y_U)$, where $y_F$ and $y_U$ are the values of $y$ typical of the folded and unfolded states. Data were fitted to a two-state model with the software ProFit (6.1) as described (*Grimsley et al., 2013*), assuming a linear urea $[D]$ dependence of $\Delta G$: $\Delta G_{U-F}^{D} = \Delta G_{U-F}^{H2O} - m[D]$, where $\Delta G_{U-F}^{D}$ is the free energy change at a given denaturant concentration, $\Delta G_{U-F}^{H2O}$ the free energy change in the absence of denaturant, and $m$ the sensitivity of the transition to denaturant. Fragment sizes of M4N were determined by ESI-micrOTOF mass spectrometry (Bruker Daltonics, Max Planck Institute core facility Martinsried), followed by bioinformatic analysis using the Find-Pept tool (ExPASy).

## Crystallization, structure solution and refinement

For crystallization, the M4N and M4NΔC protein solutions were concentrated to 70 and 30 mg/ml, respectively, in buffer A. The buffer for M4NΔC additionally contained 0.5 mM EDTA. Crystallization trials were performed at 295 K in 96-well sitting-drop vapor-diffusion plates with 50 μl of reservoir solution and drops consisting of 300 nl protein solution and 300 nl reservoir solution in the case of M4N, and 400 nl protein solution and 200 nl reservoir solution in the case of M4NΔC. Crystallization conditions for the crystals used in the diffraction experiments are listed in *Supplementary file 1H* together with the solutions used for cryo-protection. Single crystals were transferred into a droplet of cryo-protectant before loop-mounting and flash-cooling in liquid nitrogen. For experimental phasing, crystals of M4N were soaked overnight in a droplet containing reservoir solution supplemented with 5 mM $K_2PtCl_4$ prior to cryo-protection and flash-cooling. All data were collected at beamline X10SA (PXII) at the Swiss Light Source (Paul Scherrer Institute, Villigen, Switzerland) at 100 K using a PILATUS 6M detector (DECTRIS) at the wavelengths indicated in *Supplementary file 1H*. Diffraction images were processed and scaled using the XDS program suite (*Kabsch, 1993*). Using SHELXD (*Sheldrick, 2008*), three strong Pt-sites were identified in the M4N derivative dataset. After density

modification with SHELXE, the resulting electron density map could be traced by Buccaneer (*Cowtan, 2006*) to large extents, and revealed three chains of M4N in the asymmetric unit (ASU), organized as one dimer and one monomer. Refinement was continued using the native dataset. The two different crystal forms of M4NΔC, CF I and CF II, were solved by molecular replacement on the basis of the refined M4N coordinates. Using MOLREP (*Vagin and Teplyakov, 2000*), two copies of the dimeric assembly of the M4N structure were located in the ASU of CF I, and one monomer in the ASU of CF II. All models were completed by cyclic manual modeling with Coot (*Emsley and Cowtan, 2004*) and refinement with PHENIX (RRID:SCR_014224) (*Adams et al., 2010*). Analysis with PROCHECK (*Laskowski et al., 1993*) showed excellent geometries for all structures. Data collection and refinement statistics are summarized in *Supplementary file 1H*. The three structures are deposited in the PDB (*Berman et al., 2000*) with accession codes: 5FZQ (M4N), 5FZR (M4NΔC CF I), 5FZS (M4NΔC CF II).

## Testing mutations in *T. thermophilus*

*T. thermophilus* HB8 and *T. aquaticus* YT-1 were obtained from the German Collection of Microorganisms and Cell Cultures (DSMZ). Growth in liquid medium was performed under mild stirring (160 rpm) in long necked flasks at 68°C with DSMZ Medium 74 for *T. thermophilus* and DSMZ Medium 878 for *T. aquaticus*. Agar (1.6% w/v) was added to the medium for growth on plates. When required, kanamycin (30 μg/ml) and bleocin (10 μg/ml) were added to the media. For purification experiments 25 ml cultures were grown to an optic density of 0.7 $OD_{600}$ (~12 hr) and then re-inoculated in the same volume to an optical density of 0.035 $OD_{600}$. The process was repeated serially three times and two 5 ml samples were taken in each step for glycerol stocks and DNA purification. Transformation of *T. thermophilus* was performed as described previously (*Nguyen and Silberg, 2010*). Genomic and plasmid DNA from Thermus were purified from 5 ml cultures using the QIAamp DNA Mini Kit and the QIAprep Spin Miniprep Kit, respectively.

*T. thermophilus* KM4 strain was generated by gene replacement as follows: two PCR products comprising each one 1 Kb of DNA upstream and downstream of *rpsT* were amplified from *T. thermophilus* HB8 genomic DNA and then fused by overlapping PCR. The resulting fragment, in which *rpsT* is substituted by a PstI site, was cloned in the KpnI/XbaI sites of plasmid pBlueScript II SK (+). Next, a fragment from plasmid pKT1 (Biotools, Spain), which contains the thermostable kanamycin resistance *Kat* gene under the control of the constitutive PslpA promoter, was inserted into the new PstI site. Direction of the *Kat* cassette insertion was selected, so transcription from the PslpA promoter continues through *thx*, a gene that is located downstream and is predicted to form an operon with *rpsT*. The 3 Kb final construct cloned in pBluescript was subsequently amplified by PCR and the linear product was purified and transformed by electroporation in *T. thermophilus* HB8. Integration of the *Kat* cassette was selected by growth in kanamycin.

For the complementation in trans of *rpsT* from *T. thermophilus*, a PCR product of *rpsT* was amplified from genomic DNA and cloned in the SpeI/PstI sites of plasmid pJJSpro (*Nguyen and Silberg, 2010*) generating plasmid pJJSpro-rpsTTt. The same approach was followed for *rpsT* in *T. aquaticus* (pJJSpro-rpsTTa) and in *T. aquaticus rpsT* alleles with two (pJJSpro-rpsTTaM2) and four (pJJSpro-rpsTTaM4N) amino-acid substitutions. The PCR product for the two later constructs was amplified using the plasmids in which the synthesized genes were delivered as a template.

## Acknowledgements

We thank Elisabeth Weyher from the Core Facility of the MPI for Biochemistry, Martinsried, for analyzing proteins by mass spectrometry. We are grateful to the staff of beamline PXII/Swiss Light Source for their technical support. We also thank Birte Höcker, Vikram Alva and Sergey Samsonov for many helpful comments and discussions. This work was supported by institutional funds of the Max Planck Society.

# Additional information

### Funding

| Funder | Author |
| --- | --- |
| Max-Planck-Gesellschaft | Hongbo Zhu |
| | Edgardo Sepulveda |
| | Marcus D Hartmann |
| | Manjunatha Kogenaru |
| | Astrid Ursinus |
| | Eva Sulz |
| | Reinhard Albrecht |
| | Murray Coles |
| | Jörg Martin |
| | Andrei N Lupas |

The funders had no role in study design, data collection and interpretation, or the decision to submit the work for publication.

### Author contributions

HZ, ANL, Conception and design, Acquisition of data, Analysis and interpretation of data, Drafting or revising the article; ESe, MDH, JM, Acquisition of data, Analysis and interpretation of data, Drafting or revising the article; MK, Acquisition of data, Analysis and interpretation of data; AU, ESu, Acquisition of data; RA, Acquisition of data, Drafting or revising the article; MC, Analysis and interpretation of data, Drafting or revising the article

### Author ORCIDs

Edgardo Sepulveda, http://orcid.org/0000-0002-2413-8261
Marcus D Hartmann, http://orcid.org/0000-0001-6937-5677
Manjunatha Kogenaru, http://orcid.org/0000-0001-6570-7857
Andrei N Lupas, http://orcid.org/0000-0002-1959-4836

# Additional files

### Supplementary files

• Supplementary file 1. Further supporting computational and experimental results. (A) Sequence variation in RPS20-hh at positions 6, 7, 9 and 23 (TPR unit numbering) observed in RPS20 sequences. (B) Most commonly observed amino acids in RPS20-hh. (C) List of putative TPR homologs identified in the PDB by sequence and structure analysis. (D) RPS20-hh sequences that resemble a TPR profile according to TPRpred. (E) Mutations tested in silico on RPS20-hh for TPR design. (F) Biophysical parameters of designed TPRs. (G) Primary structures of M4N molecules observed in the crystal structures. (H) Crystallization conditions, and data collection/refinement statistics. (I) Detailed structure comparison results of different chains in M4N structures, and of M4N to CTPR3. (J) SEG prediction of low-complexity regions in RPS20-hhta.

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
