## [Decision Letter]

Thank you for submitting your article "Origin of a folded repeat protein from an intrinsically unfolded ancestor" for consideration by *eLife*. Your article has been favorably evaluated by John Kuriyan as the Senior Editor and three reviewers, including Dan S. Tawfik (Reviewer #3) and Nir Ben-Tal (Reviewer #1), who is a member of our Board of Reviewing Editors.

The reviewers have discussed the reviews with one another and the Reviewing Editor has drafted this decision to help you prepare a revised submission.

Summary:

The lead author, Lupas, has suggested back in 2001 a model for the emergence of proteins in our evolutionary history. According to his model, short amino acid fragments that were capable of RNA binding emerged. Of these, some improved RNA function, and survived. Eventually, some of these became foldable (and functional) on their own. Proteins emerged by duplications of these ancestral fragments. In an *eLife* publication last year, the Lupas lab conducted bioinformatics analysis and detected a gallery of 40 commonly found protein fragments. That they are popular, and found also in ancient proteins, makes them candidates as ancestral fragments. Here, Lupas and colleagues examine one of these: an α helical hairpin with about 35-residues long sequence motif, known as tetratricopeptide. Sequence analysis showed that it appears also in the ribosomal protein RPS20, but it is not stable on its own, detached from the ribosome. Using a triplet of this motif as a seed, the authors managed to design variants that are foldable autonomously. Because they are distant only 2-5 mutations (per motif) from the original, the authors suggest that this is a viable evolutionary scenario.

This work is new and interesting in three respects. Firstly, there have been several demonstrations of the general principle of peptides leading to globular proteins via duplication and fusion, but not of the fold addressed here (TPRs). Secondly, the peptide's origin was identified here, and assigned to an omnipresent, ancient protein (a ribosomal protein) – in previous works, the ancestral peptide has been reconstructed, but its origins remain unknown. Thirdly, the ancestral peptide has no structure on its own, and assumes the TPR fold upon duplication and fusion (in previous demonstrations, the peptide precursors oligomerized to give a globular protein that resembles the modern, monomeric repeat protein). The work also provides a further, and interesting, demonstration of the principle that mutations that are neutral in one context can pave the road to new trait, or structures in this case. Comments for further improvements are provided below.

Essential revisions:

1) The title and emphasis on "intrinsically unfolded" is misleading and may be misinterpreted by causal readers. The authors are not proposing that globular proteins originated from low complexity stretches of protein sequence. Although a single α-hairpin is indeed unfolded without RNA because it is simply too short, it does have a good α-helical content, and indeed is an α-hairpin rather than low complexity intrinsically unfolded protein.

2) The constructed protein had been shown to adopt the TPR fold, but has no biochemical function, and as we know, structure in itself does is not 'visible' by natural selection. Presumably identifying potential function(s) for these designed TPRs is not a trivial task. However, the authors should explicitly acknowledge this and discuss what might be the function(s) which this protein could perform.

3) A related issue: The likelihood of an evolutionary step that involves 2-5 mutations should be discussed further. Reassuringly, none of the mutations harm the parent organism (Thermus has been used here), but would such mutants survive without providing evolutionary value? Or would they provide some merit even alone? Perhaps still bound to RNA? What is the evolutionary scenario here?

---

## [Author Response]

Summary:

*The lead author, Lupas, has suggested back in 2001 a model for the emergence of proteins in our evolutionary history. According to his model, short amino acid fragments that were capable of RNA binding emerged. Of these, some improved RNA function, and survived. Eventually, some of these became foldable (and functional) on their own. Proteins emerged by duplications of these ancestral fragments. In an eLife publication last year, the Lupas lab conducted bioinformatics analysis and detected a gallery of 40 commonly found protein fragments. That they are popular, and found also in ancient proteins, makes them candidates as ancestral fragments. Here, Lupas and colleagues examine one of these: an α helical hairpin with about 35-residues long sequence motif, known as tetratricopeptide. Sequence analysis showed that it appears also in the ribosomal protein RPS20, but it is not stable on its own, detached from the ribosome. Using a triplet of this motif as a seed, the authors managed to design variants that are foldable autonomously. Because they are distant only 2-5 mutations (per motif) from the original, the authors suggest that this is a viable evolutionary scenario.*

*This work is new and interesting in three respects. Firstly, there have been several demonstrations of the general principle of peptides leading to globular proteins via duplication and fusion, but not of the fold addressed here (TPRs). Secondly, the peptide's origin was identified here, and assigned to an omnipresent, ancient protein (a ribosomal protein) – in previous works, the ancestral peptide has been reconstructed, but its origins remain unknown. Thirdly, the ancestral peptide has no structure on its own, and assumes the TPR fold upon duplication and fusion (in previous demonstrations, the peptide precursors oligomerized to give a globular protein that resembles the modern, monomeric repeat protein). The work also provides a further, and interesting, demonstration of the principle that mutations that are neutral in one context can pave the road to new trait, or structures in this case. Comments for further improvements are provided below.*

We now realize, based on the summary and the reviewer comments below, that we had not formulated the basis of our study with sufficient clarity, and have therefore made multiple changes throughout the manuscript in revision. We did not wish to imply a scenario where RPS20 was the ancestor of the TPR fold, but rather one of common descent of RPS20 and TPR from an ancestral hairpin, which also gave rise to a number of other folds. For our study, we used RPS20 because we consider it the oldest still surviving descendant of this ancestral hairpin and, due to its slow rate of evolution, also the one most likely to reflect ancestral properties. In the process, we hoped to find that repetition is a sufficiently powerful mechanism to allow the emergence of a folded protein from an unstructured parent (note that the entire RPS20 is unstructured in the absence of ribosomal RNA, not just the isolated helix hairpin).

Essential revisions:

*1) The title and emphasis on "intrinsically unfolded" is misleading and may be misinterpreted by causal readers. The authors are not proposing that globular proteins originated from low complexity stretches of protein sequence. Although a single α-hairpin is indeed unfolded without RNA because it is simply too short, it does have a good α-helical content, and indeed is an α-hairpin rather than low complexity intrinsically unfolded protein.*

We agree that the term will be misunderstood by readers who consider that only proteins unstructured under all conditions are in fact intrinsically disordered. However, it has become widely accepted that unstructured proteins that become partly or fully structured upon binding to small molecules or macromolecular ligands are also usefully included among the intrinsically disordered proteins. We have changed the title to use the term “disordered”, which is the generally used term (rather than “unfolded”, which we had used before as a contrast to the “folded” repeat protein) and have added a sentence to the Introduction to explain why we refer to ribosomal proteins as intrinsically disordered, also providing references to recent high-profile reviews that can further illuminate this point. We have further added material to section 2.4 and the Methods in order to substantiate that the sequence of RPS20 is in fact similar to that of intrinsically disordered proteins as judged by many disorder predictors, and with a biased residue distribution seen as low-complexity by the program SEG.

With respect to the single α-hairpin being unfolded, we agree that its lack of structure in solution would not be surprising, but that was not the point we were trying to make. Our point was that the parent protein is unstructured in solution. For this reason, we deleted “intrinsically unfolded” from the description of the helical hairpin in the last sentence of the Abstract.

*2) The constructed protein had been shown to adopt the TPR fold, but has no biochemical function, and as we know, structure in itself does is not 'visible' by natural selection. Presumably identifying potential function(s) for these designed TPRs is not a trivial task. However, the authors should explicitly acknowledge this and discuss what might be the function(s) which this protein could perform.*

We fully agree with this point and have expanded the last section of the paper to explicitly address it. Structure is a prerequisite for function, but it is the function that is under selection. Although the goal of our study was to explore the origin of structure from an unstructured precursor, we had been aware all along that the survival of this newly emerged structure would be dependent on the benefits it confers.

Thus, we started experiments on this point as soon as we had obtained the M4N structure. Following the reviewers’ comments, we decided to extend these in revision, but have not succeeded in obtaining conclusive results so far.

Guided by the observation that TPR domains, where of known activity, mediate protein-protein interactions, we started screens to identify potential binders to our new protein. Specifically, we performed pulldowns with soluble cell extracts from *T. aquaticus, T. thermophilus* and *E. coli* using M4NΔC and M5ΔC, but were unable to discover interactors of even moderate affinity. Since TPR domains typically recognize linear sequence motifs in unstructured segments, we also screened a peptide library against M4NΔC and M5ΔC, using phage display with the Ph.D.-12 Library (NEB) containing ca. 10_9_ 12-mers, but again, so far, no peptides could be identified (these experiments are still ongoing). This inability to identify protein or peptide ligands binding to our TPR domains may well be due to the dimerization of the protein, which obstructs the ligand-binding face common to TPR proteins. We are therefore now exploring whether we can disrupt dimerization by single mutations in the C-terminal helices before resuming these screens.

In parallel, however, we are also trying to exploit the central crevice formed in the dimer to find small- molecule ligands that could open an entirely new set of functionalities not seen in TPR proteins today.

Finally, we are exploring to what extent our new proteins still retain the ability of the RPS20 parent to interact with RNA. Preliminary experiments show that the folded dimers have no affinity for heterologous RNA molecules, but experiments remain to be done with Thermus ribosomal RNA and with the yet to be obtained monomeric variants.

*3) A related issue: The likelihood of an evolutionary step that involves 2-5 mutations should be discussed further. Reassuringly, none of the mutations harm the parent organism (Thermus has been used here), but would such mutants survive without providing evolutionary value? Or would they provide some merit even alone? Perhaps still bound to RNA? What is the evolutionary scenario here?*

We agree that the evolutionary scenario did not become clear enough from the manuscript and we therefore made multiple changes to clarify it. We did not wish to imply that TPR proteins arose from a proto-RPS20 in a fully formed cell (such as Thermus is today); rather, we think that they arose from a precursor form that, like RPS20, interacted with RNA, but in a much more primitive cell-like environment. The reason for the in vivoexperiments was to establish that the mutations necessary for folding are neutral with respect to RNA interaction in our model system. We take this finding to suggest that these (and other fold-promoting) mutations could have been sampled by neutral drift also in the ancestral hairpin. Indeed, it is reasonable to expect that in a more primitive, less networked and less integrated setting, many more mutations would have been (largely) neutral than in the Thermus cytosol. A further corollary of the evolutionary steps happening in a pre-cellular environment is that, in the absence of performant control and repair mechanisms, the frequency of all kinds of genetic mutations would have been much higher.